# Risk factors affecting polygenic score performance across diverse cohorts

Daniel Hui[1], Scott Dudek[1], Krzysztof Kiryluk[2], Theresa L Walunas[3], Iftikhar J Kullo[4], Wei-Qi Wei[5], Hemant Tiwari[6], Josh F Peterson[5], Wendy K Chung[7], Brittney H Davis[8], Atlas Khan[2], Leah C Kottyan[9], Nita A Limdi[8], Qiping Feng[10], Megan J Puckelwartz[11], Chunhua Weng[12], Johanna L Smith[4], Elizabeth W Karlson[13], Regeneron Genetics Center, Penn Medicine BioBank, Gail P Jarvik[14], Marylyn D Ritchie[1]*

[1]Department of Genetics, Perelman School of Medicine, University of Pennsylvania, Philadelphia, United States; [2]Division of Nephrology, Department of Medicine, Columbia University, New York, United States; [3]Department of Preventive Medicine, Northwestern University Feinberg School of Medicine, Chicago, United States; [4]Department of Cardiovascular Medicine, Mayo Clinic, Rochester, United States; [5]Department of Biomedical Informatics, Vanderbilt University Medical Center, Nashville, United States; [6]Department of Pediatrics, University of Alabama at Birmingham, Birmingham, United States; [7]Departments of Pediatrics and Medicine, Columbia University Irving Medical Center, Columbia University, New York, United States; [8]Department of Neurology, School of Medicine, University of Alabama at Birmingham, Birmingham, United States; [9]The Center for Autoimmune Genomics and Etiology, Division of Human Genetics, Cincinnati Children's Hospital Medical Center, Cincinnati, United States; [10]Division of Clinical Pharmacology, Department of Medicine, Vanderbilt University Medical Center, Nashville, United States; [11]Center for Genetic Medicine, Northwestern University Feinberg School of Medicine, Chicago, United States; [12]Department of Biomedical Informatics, Vagelos College of Physicians & Surgeons, Columbia University, New York, United States; [13]Division of Rheumatology, Inflammation, and Immunity, Department of Medicine, Brigham and Women's Hospital and Harvard Medical School, Boston, United States; [14]Departments of Medicine (Medical Genetics) and Genome Sciences, University of Washington Medical Center, Seattle, United States

*For correspondence: marylyn@pennmedicine.upenn.edu

## eLife assessment

This study presents a **convincing** analysis of the effects of covariates, such as age, sex, socioeconomic status, or biomarker levels, on the predictive accuracy of polygenic scores for body mass index; the work is further supported by **important** approaches for improving prediction accuracy by accounting for such covariates across a variety of association studies. The authors did a commendable job addressing reviewer suggestions and comments. The work will be of interest to colleagues using and developing methods for phenotypic prediction based on polygenic scores.

**Abstract** Apart from ancestry, personal or environmental covariates may contribute to differences in polygenic score (PGS) performance. We analyzed the effects of covariate stratification and interaction on body mass index (BMI) PGS ($PGS_{BMI}$) across four cohorts of European (N = 491,111) and African (N = 21,612) ancestry. Stratifying on binary covariates and quintiles for continuous

covariates, 18/62 covariates had significant and replicable $R^2$ differences among strata. Covariates with the largest differences included age, sex, blood lipids, physical activity, and alcohol consumption, with $R^2$ being nearly double between best- and worst-performing quintiles for certain covariates. Twenty-eight covariates had significant $PGS_{BMI}$–covariate interaction effects, modifying $PGS_{BMI}$ effects by nearly 20% per standard deviation change. We observed overlap between covariates that had significant $R^2$ differences among strata and interaction effects – across all covariates, their main effects on BMI were correlated with their maximum $R^2$ differences and interaction effects (0.56 and 0.58, respectively), suggesting high-$PGS_{BMI}$ individuals have highest $R^2$ and increase in PGS effect. Using quantile regression, we show the effect of $PGS_{BMI}$ increases as BMI itself increases, and that these differences in effects are directly related to differences in $R^2$ when stratifying by different covariates. Given significant and replicable evidence for context-specific $PGS_{BMI}$ performance and effects, we investigated ways to increase model performance taking into account nonlinear effects. Machine learning models (neural networks) increased relative model $R^2$ (mean 23%) across datasets. Finally, creating $PGS_{BMI}$ directly from GxAge genome-wide association studies effects increased relative $R^2$ by 7.8%. These results demonstrate that certain covariates, especially those most associated with BMI, significantly affect both $PGS_{BMI}$ performance and effects across diverse cohorts and ancestries, and we provide avenues to improve model performance that consider these effects.

## Introduction

Polygenic scores (PGS) provide individualized genetic predictors of a phenotype by aggregating genetic effects across hundreds or thousands of loci, typically estimated from genome-wide association studies (GWAS). In recent years, it has become increasingly apparent that the transferability of PGS performance across different cohorts is poor (*Martin et al., 2019*). Most analyses to date have focused on ancestry differences as the main driver of this lack of portability (*Wang et al., 2020*; *Galinsky et al., 2019*; *Shi et al., 2021*). However, a growing body of evidence has demonstrated that PGS performance and effect estimates are influenced by differences in certain contexts, that is, environmental (classically termed 'gene–environment' effects or interactions) or personal-level covariates – different phenotypes seem to be differently affected by these covariates, with adiposity traits such as body mass index (BMI) having substantial evidence for these effects (*Rask-Andersen et al., 2017*; *Robinson et al., 2017*; *Sulc et al., 2020*; *Justice et al., 2017*; *Helgeland et al., 2019*; *Vogelezang et al., 2020*; *Couto Alves et al., 2019*; *Choh et al., 2014*; *Mostafavi et al., 2020*; *Elks et al., 2012*). In one previous study, they showed that GWAS stratified by sample characteristics had better PGS performance in cohorts that matched the sample characteristics of the stratified GWAS, and that differences in heritability between the stratified cohorts partially explained this observation (*Mostafavi et al., 2020*).

There are several gaps in current knowledge about these covariate-specific effects. Many analyses have assessed only a handful of these covariates due to the myriad of choices possible in typical large-scale biobanks. Little investigation has been done to systematically understand why certain covariates affect PGS performance, with such knowledge being useful to reduce the potential search for variables that impart context-specific effects. Furthermore, most studies investigating PGS–covariate interactions have been in European ancestry individuals; notably, comparing differences in PGS performance and prediction while controlling for differences in ancestry versus differences in context has not been assessed in previous studies. Moreover, covariate-specific effects are notorious for replicating poorly in human genetics studies, and previous studies of PGS–covariate interactions have been predominantly performed in the UK Biobank (UKBB) (*Bycroft et al., 2018*), where the majority of individuals are aged 40–69 (i.e., excluding young adults), are overall healthier than those from other, for example, hospital-based cohorts, and are predominantly European ancestry. Additionally, PGS performance is often assessed using linear models and in isolation of clinical covariates, which in practice would often be available. Machine learning models can have increased performance over linear models and are capable of modeling complex relationships and interactions between variables, which may serve to increase predictive performance, especially given evidence for PGS–covariate-specific effects. Finally, given evidence for context-specific effects, it should be possible to directly incorporate SNP–covariate interaction effects from a GWAS directly to improve prediction performance, instead of relying on post hoc interactions from a typical PGS calculated from main GWAS effects.

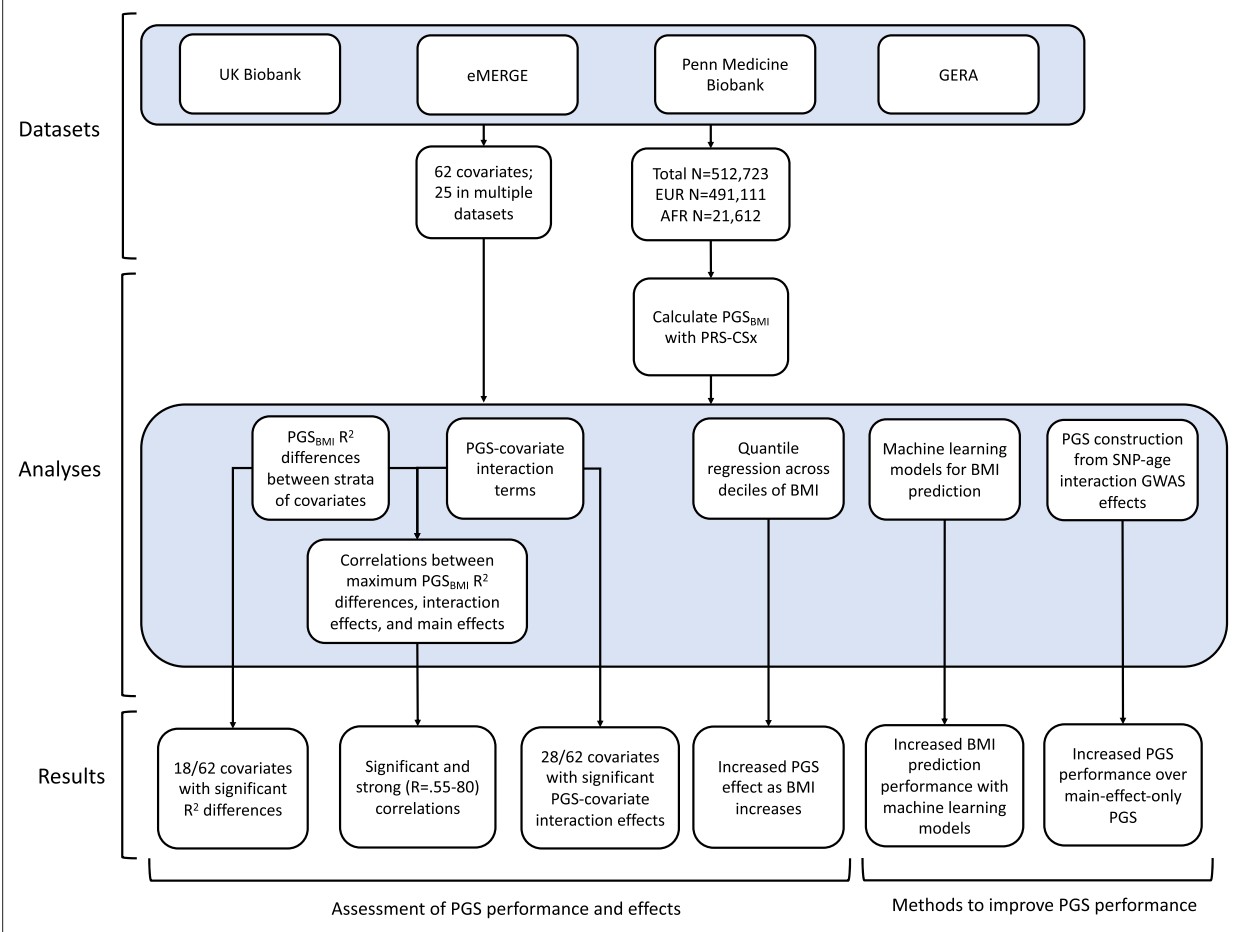

**Figure 1.** A flowchart of the project.

Using genetic data with linked-phenotypic information from electronic health records, we estimated the effects of covariate stratification and interaction on performance and effect estimates of PGS for BMI (PGS$_{BMI}$) – a flowchart summarizing our analyses is presented in *Figure 1*. These analyses were done across four datasets (*Supplementary file 1a*): UKBB, Penn Medicine BioBank (PMBB) (*Verma et al., 2022*), Electronic Medical Records and Genomics (eMERGE) network dataset (*Stanaway et al., 2019*), and Genetic Epidemiology Research on Adult Health and Aging (GERA) (*Banda et al., 2015*). These datasets include participants from two ancestry groups (N = 491,111 European ancestry [EUR], N = 21,612 African ancestry [AFR]), and 62 covariates (25 present in multiple datasets) representing laboratory, survey, and biometric data types typically associated with cardiometabolic health and adiposity. After constructing PGS$_{BMI}$ using out-of-sample multi-ancestry BMI GWAS, we assessed the effects of covariate stratification on PGS$_{BMI}$ $R^2$, the significance of PGS$_{BMI}$–covariate interaction terms and their increases to model $R^2$ over models only using main effects, as well as correlation of main effect, interaction effect, and $R^2$ differences. We then assessed ways to increase model performance through using machine learning models, and creating PGS$_{BMI}$ using GxAge GWAS effects. This study addresses a plethora of open issues considering performance and effects of PGS on individuals from diverse backgrounds.

## Results
### Effect of covariate stratification on PGS$_{BMI}$ performance

We assessed 62 covariates for PGS$_{BMI}$ $R^2$ differences (25 present, or suitable proxies, in multiple datasets *Supplementary file 1b*) after stratifying on binary covariates and quintiles for continuous covariates. With UKBB EUR as discovery (N = 376,729), 18 covariates had significant differences (Bonferroni

p<0.05/62) in $R^2$ among groups (*Figure 2a*), including age, sex, alcohol consumption, different physical activity measurements, Townsend deprivation index, different dietary measurements, lipids, blood pressure, and HbA1c, with 40 covariates having suggestive (p<0.05) evidence of $R^2$ differences. From an original $PGS_{BMI}$ $R^2$ of 0.076, $R^2$ increased to 0.094–0.088 for those in the bottom physical activity, alcohol intake, and high-density lipoprotein (HDL) cholesterol quintiles, and decreased to 0.067–0.049 for those in the top quintile, respectively, comparable to differences observed between ancestries (*Martin et al., 2019*). We note that the differences in $R^2$ due to alcohol intake and HDL were larger than those of any physical activity phenotype, despite physical activity having one of the oldest and most replicable evidence of interaction with genetic effects of BMI (*Kilpeläinen et al., 2011*; *Rampersaud et al., 2008*). Despite considerable published evidence suggesting covariate-specific genetic effects between BMI and smoking behaviors (*Robinson et al., 2017*; *Justice et al., 2017*), we were only able to find suggestive evidence for $R^2$ differences when stratifying individuals across several smoking phenotypes (minimum p=0.016, for smoking pack years). $R^2$ differences due to educational attainment were also only suggestive (p=0.015), with published evidence on this association being conflicting (*Amin et al., 2018*; *Li et al., 2021*; *Frank et al., 2019*).

We replicated these analyses in three additional large-scale cohorts of European and African ancestry individuals (*Figure 2b*, *Supplementary file 1c*), as well as in African ancestry UKBB individuals. Among covariates with significant performance differences in the discovery analysis, we were able to replicate significant (p<0.05) $R^2$ differences for age, HDL cholesterol, alcohol intake frequency, physical activity, and HbA1c, despite much smaller sample sizes. We again observed mostly insignificant differences across cohorts and ancestries when stratifying due to different smoking phenotypes and educational attainment. For each covariate and ancestry combination, we combined data across cohorts and conducted a linear regression weighted by sample size, regressing $R^2$ values on covariate values across groupings. Slopes of the regressions across cohorts had different signs between ancestries for the same covariate (triglyceride levels, HbA1c, diastolic blood pressure, and sex), although larger sample sizes may be needed to confirm these differences are statistically significant.

Several observations related to age-specific effects on $PGS_{BMI}$ we considered noteworthy. First, in the weighted linear regression of all $R^2$ values across ancestries, expected $R^2$ for African ancestry individuals can become greater than that of European ancestry individuals among individuals within bottom and top age quintiles observed in these data. For instance, the predicted $R^2$ of 0.048 for 80-year-old European ancestry individuals would be lower than that of African ancestry individuals aged 24.7 and lower, indicating that differences in covariates can affect $PGS_{BMI}$ performance more than differences due to ancestry. Second, we obtained these results despite the average age of GWAS individuals being 57.8, which should increase $PGS_{BMI}$ $R^2$ for individuals closest to this age (*Mostafavi et al., 2020*). This result suggests that PGS performance due to decreased heritability with age cannot be fully reconciled using GWAS from individuals of similar age being used to create $PGS_{BMI}$ (as heritability is an upper bound on PGS performance). Finally, we observed that $PGS_{BMI}$ $R^2$ increases as age decreases, consistent with published evidence suggesting that the heritability of BMI decreases with age (*Ge et al., 2017*; *Min et al., 2013*).

## PGS–covariate interaction effects

Next, we estimated the differences in PGS effects due to interactions with covariates by modeling interaction terms between $PGS_{BMI}$ and the covariate for each covariate in our list (described in 'Materials and methods'). We implemented a correction for shared heritability between covariates of interest and outcome (which can inflate test statistics *Aschard et al., 2015*) to better measure the environmental component of each covariate, and show that this correction successfully reduces significance of interaction estimates (*Figure 3—figure supplement 1*). Again, using UKBB EUR as the discovery cohort, we observed 28 covariates with significant (Bonferroni p<0.05/62) PGS–covariate interactions (*Table 1*), with 38 having suggestive (p<0.05) evidence (*Supplementary file 1d*). We observed the largest effect of PGS–covariate interaction with alcohol drinking frequency (20.0% decrease in PGS effect per 1 standard deviation [SD] increase, p=$2.62 \times 10^{-55}$), with large effects for different physical activity measures (9.4–12.5% decrease/SD, minimum p=$3.11 \times 10^{-66}$), HDL cholesterol (15.3% decrease/SD, p=$1.71 \times 10^{-96}$), and total cholesterol (12.7% decrease/SD, p=$1.64 \times 10^{-71}$). We observed significant interactions with diastolic blood pressure (10.8% increase/SD, p=$6.06 \times 10^{-60}$), but interactions with systolic blood pressure were much smaller (1.17% increase/

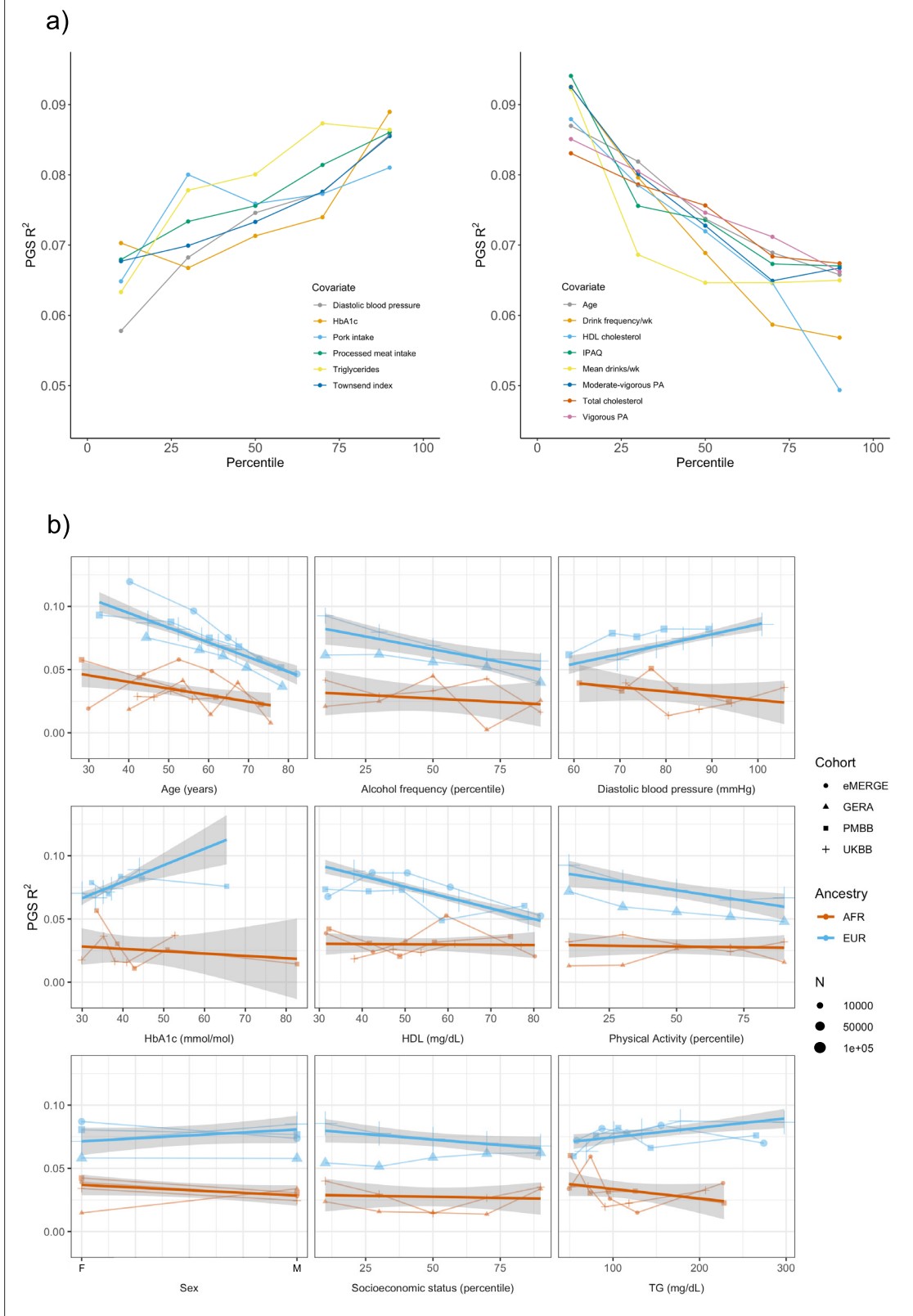

**Figure 2.** Polygenic score (PGS) $R^2$ stratified by quintiles for quantitative variables and by binary variables. (**a**) Continuous covariates with significant ($p<8.1 \times 10^{-4}$) $R^2$ differences across quintiles in UK Biobank (UKBB) European ancestry (EUR). Pork and processed meat consumption per week were excluded from this plot in favor of pork and processed meat intake. (**b**) Covariates with significant differences that were available in multiple cohorts. When traits had the same or directly comparable units between cohorts we show the actual trait values (and show percentiles for physical activity,

*Figure 2 continued on next page*

*Figure 2 continued*

alcohol intake frequency, and socioeconomic status, which had slightly differing phenotype definitions across cohorts) plotted on x-axis. Townsend index and income were used as variables for socioeconomic status in UKBB and Genetic Epidemiology Research on Adult Health and Aging (GERA), respectively. Note that the sign for Townsend index was reversed, since increasing Townsend index is lower socioeconomic status, while increasing income is higher socioeconomic status. PA, physical activity (PA); IPAQ, International Physical Activity Questionnaire.

---

SD, p=4.41 × 10$^{-3}$). Significant interactions with HbA1c (4.63% increase/SD, p=5.37 × 10$^{-14}$) and type 2 diabetes (27.2% PGS effect increase in cases, p=1.83 × 10$^{-7}$) were also observed. Other significant PGS–covariate interactions included lung function, age, sex, and LDL cholesterol – various dietary measurements also had significant interactions, albeit with smaller effects than other significant covariates. We were able to find significant interaction effects for smoking pack years (4.78% increase/SD, p=3.68 × 10$^{-7}$), but other smoking phenotypes had insignificant interaction effects after correcting for multiple tests (minimum p=2.7 × 10$^{-3}$); interactions with educational attainment were also insignificant (p=4.54 × 10$^{-2}$).

We replicated these analyses across ancestries and the other non-UKBB EUR cohorts (*Figure 3*, *Supplementary file 1d*). For age and sex, which were available for all cohorts, interactions were significant (p<0.05) and directionally consistent across cohorts and ancestries (except for GERA AFR, which had small sample size [N = 1,789]). We were able to test interactions with alcohol intake frequency and physical activity in GERA, and replicated significant and directionally consistent associations. We observed poor replication for LDL cholesterol, HbA1c, and smoking pack years, with insignificant and directionally inconsistent interaction effects across cohorts. Educational attainment was available in GERA, and interactions were once again insignificant. We observed significant and directionally consistent interaction effects for TG in eMERGE EUR and PMBB EUR, while the effect was inconsistent in UKBB EUR despite much larger sample size.

However, despite significance of interaction terms, increases in model R$^2$ when including PGS–covariate interaction terms were small. For instance, the maximum increase among all covariates in UKBB EUR was only 0.0024 from a base R$^2$ of 0.1049 (2.1% relative increase), for alcoholic drinks per week. Across all cohorts and ancestries, the maximum increase in R$^2$ was only 0.0058 from a base R$^2$ of 0.09454 (6.1% relative increase), when adding a PGS–age interaction term for eMERGE EUR (p=5.40 × 10$^{-46}$) – this was also the largest relative increase among models with significant interaction terms. This result suggests that, while interaction effects can significantly modify PGS$_{BMI}$ effect, their overall impact on model performance is relatively small, despite large differences in R$^2$ when stratifying by covariates.

## Correlations between R$^2$ differences, interaction effects, and main effects

We next investigated the relationship between interaction effects, maximum R$^2$ differences across quintiles, and main effects of covariates on BMI. We first estimated the main effects of each covariate on BMI ('Materials and methods', *Supplementary file 1e*), and then calculated the correlation weighted by sample size between main effects, maximum PGS$_{BMI}$ R$^2$ across quintiles, and PGS–covariate interaction effects (*Figure 4*) across all cohorts and ancestries – GERA data were excluded from these analyses due to slightly different phenotype definition (*Supplementary file 1f*), as were binary variables. Interaction effects and maximum R$^2$ differences had a 0.80 correlation (p=2.1 × 10$^{-27}$), indicating that variables with larger interaction effects also had larger effects on PGS$_{BMI}$ performance across quintiles, and that covariates that increase PGS$_{BMI}$ effect also have the largest effect on PGS$_{BMI}$ performance, that is, individuals most at risk for obesity will have both disproportionately larger PGS$_{BMI}$ effect and R$^2$. Main effects and maximum R$^2$ differences had a 0.56 correlation (p=1.3 × 10$^{-11}$), while main effects and interaction effects had a 0.58 correlation (p=7.6 × 10$^{-12}$), again suggesting that PGS$_{BMI}$ are more predictive in individuals with higher values of BMI-associated covariates, although less predictive than estimating the interaction effects themselves directly. However, this result demonstrates that covariates that influence both PGS$_{BMI}$ effect and performance can be predicted just using main effects of covariates, which are often known for certain phenotypes and easier to calculate, as genetic data and PGS construction would not be required.

**Table 1.** Model descriptive statistics on 28 of 62 covariates, which have significant (p<0.05/62) polygenic score (PGS)–covariate interaction terms, in UK Biobank (UKBB) European ancestry (EUR).

The third column is the percentage change in PGS effect per unit change (standard deviations for continuous variables, binary variables encoded as 0 or 1) in covariate. The fifth column is the increase in model $R^2$ with a PGS–covariate interaction term versus a main effects only model.

| Variable type | Covariate | % change in $\beta_{PGS}$ per covariate unit change | Interaction p | | $R^2$ increase with interaction term | N |
|---|---|---|---|---|---|---|
| | HDL cholesterol | −15.29 | $1.71 \times 10^{-96}$ | 0.0012 | | 328,719 |
| | Total cholesterol | −12.70 | $1.64 \times 10^{-71}$ | 0.00082 | | 359,221 |
| | IPAQ | −12.50 | $3.11 \times 10^{-66}$ | 0.001 | | 304,951 |
| | Moderate-vigorous PA | −11.41 | $8.92 \times 10^{-65}$ | 0.001 | | 304,951 |
| | Diastolic BP | 10.84 | $6.06 \times 10^{-60}$ | 0.0007 | | 352,804 |
| | Townsend Index | 6.78 | $2.86 \times 10^{-58}$ | 0.00089 | | 376,283 |
| | Age | −9.02 | $3.60 \times 10^{-57}$ | 0.00061 | | 376,729 |
| | FVC | −9.66 | $4.69 \times 10^{-56}$ | 0.0008 | | 343,467 |
| | Drink frequency/week | −19.96 | $2.62 \times 10^{-55}$ | 0.0024 | | 122,281 |
| | LDL cholesterol | −9.86 | $2.63 \times 10^{-51}$ | 0.00058 | | 358,556 |
| | N days vigorous PA/week | −9.37 | $2.42 \times 10^{-35}$ | 0.0007 | | 299,963 |
| | FEV1 | −7.38 | $7.15 \times 10^{-35}$ | 0.0005 | | 343,544 |
| | Mean alcohol consumption | −7.38 | $7.65 \times 10^{-22}$ | 0.00113 | | 126,756 |
| | HbA1c | 4.63 | $5.37 \times 10^{-14}$ | 0.0002 | | 358,798 |
| | Mean drinks/week | −7.66 | $1.01 \times 10^{-13}$ | 0.0008 | | 112,204 |
| | Water intake | 4.60 | $2.97 \times 10^{-13}$ | 0.00014 | | 347,472 |
| | Processed meat intake | 3.70 | $2.38 \times 10^{-7}$ | 0.0002 | | 376,205 |
| | Starch mean | 5.51 | $3.15 \times 10^{-7}$ | 0.00018 | | 128,346 |
| | Smoking pack years | 4.78 | $3.68 \times 10^{-7}$ | 0.0002 | | 114,135 |
| | Protein mean | 4.82 | $6.52 \times 10^{-7}$ | 0.00018 | | 128,181 |
| | Saturated fat mean | 4.92 | $1.23 \times 10^{-6}$ | 0.00017 | | 127,899 |
| | Fat mean | 4.40 | $1.64 \times 10^{-5}$ | 0.00013 | | 128,092 |
| | Saturated fat grams/week | 2.46 | $1.79 \times 10^{-5}$ | $4.00 \times 10^{-5}$ | | 364,629 |
| Continuous | Retinol mean | 3.77 | $3.54 \times 10^{-4}$ | $9.00 \times 10^{-5}$ | | 126,029 |
| | IPAQ | −12.68 | $5.30 \times 10^{-62}$ | 0.0009 | | 304,951 |
| | Vigorous PA/week | −20.55 | $9.07 \times 10^{-54}$ | 0.0009 | | 304,951 |
| | Sex | −11.02 | $1.41 \times 10^{-24}$ | 0.00025 | | 376,729 |
| Binary | Diabetes | 27.19 | $1.83 \times 10^{-7}$ | 0.0004 | | 375,903 |

BP = blood pressure, PA = physical activity, FVC = forced vital capacity, FEV1 = forced expiratory volume in 1 s, HDL = high-density lipoprotein, LDL = high-density lipoprotein, IPAQ = International Physical Activity Questionnaire.

## Increase in PGS effect for increasing percentiles of BMI itself, and its relation to $R^2$ differences when stratifying by covariates

Given large and replicable correlations between main effects, interaction effects, and maximum $R^2$ differences for individual covariates, it seemed these differences may be due to the differences in

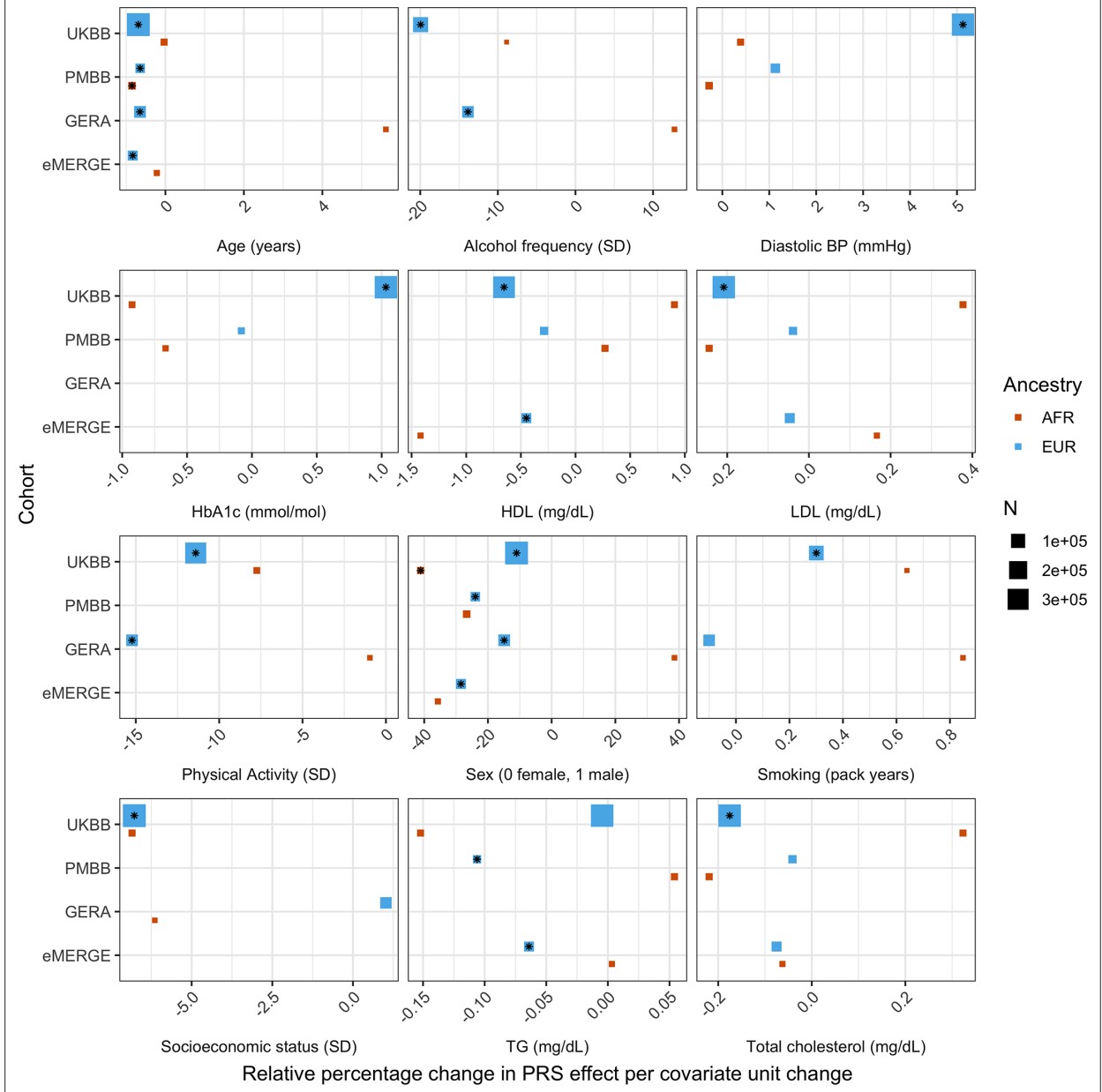

**Figure 3.** Relative percentage changes in polygenic score (PGS) effect per unit change in covariate, for covariates that significantly changed PGS effect (i.e., significant interaction beta at Bonferroni $p < 8.1 \times 10^{-4}$ – denoted by asterisks) and were present in multiple cohorts and ancestries. Same covariate groupings and transformations were performed as in *Figure 1*. Similarly, actual values were used when variables had comparable units across cohorts, and standard deviations (SD) used otherwise.

The online version of this article includes the following figure supplement(s) for figure 3:

**Figure supplement 1.** Polygenic score (PGS)–covariate interaction term $-\log_{10}$(p-values) in UK Biobank (UKBB) European ancestry (EUR), with and without including the covariate PGS in the model – the mean $-\log_{10}(p)$ is reduced from 18.0899 to 14.97072 with their inclusions.

BMI itself, rather than any individual or combination of covariates. To assess this, we used quantile regression to evaluate the effect of $PGS_{BMI}$ on BMI at different deciles of BMI itself. We observed that the effect of $PGS_{BMI}$ consistently increases from lower deciles to higher deciles across all cohorts and ancestries (*Figure 5*) – for instance, in European ancestry UKBB individuals, the effect of $PGS_{BMI}$ (in units of log(BMI)) when predicting the bottom decile of log(BMI) was 0.716 (95% CI: 0.701–0.732), and increased to 1.31 (95% CI: 1.29–1.33) in the top decile. Across all cohorts and ancestries, the effect of $PGS_{BMI}$ between lowest and highest effect decile ranged from 1.43 to 2.06 times larger, with all

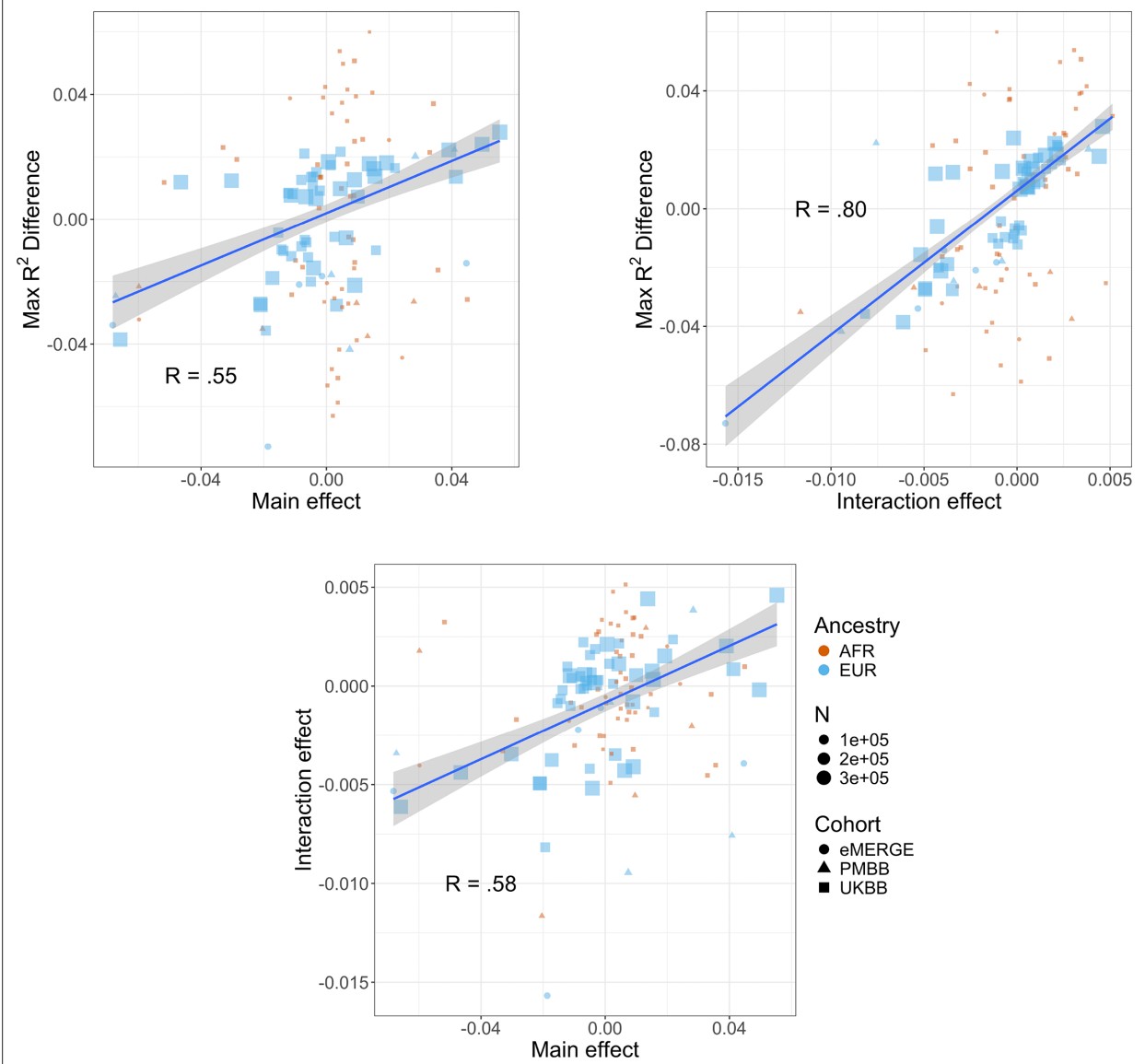

**Figure 4.** Relationships (Pearson correlations weighted by sample size) between maximum $R^2$ differences across strata, main effects of covariate on log(BMI), and polygenic score (PGS)–covariate interaction effects on log(BMI). Main effect units are in standard deviations, interaction effect units are in PGS standard deviations multiplied by covariate standard deviations. Only continuous variables are plotted and modeled. Genetic Epidemiology Research on Adult Health and Aging (GERA) was excluded due to slightly different phenotype definitions. BMI, body mass index.

cohorts and ancestries having nonoverlapping 95% confidence intervals between their effects (except for African ancestry eMERGE individuals, which had much smaller sample size).

While this analysis showed that the effect of $PGS_{BMI}$ increases as BMI itself increases, which may help explain significant interaction effects between $PGS_{BMI}$ and different covariates, it does not directly explain differences in $R^2$ when stratifying by different covariates – we describe several points that help explain this result and suggest they may actually be closely related. Essentially, as the magnitude of the slope of a regression line increases while the mean squared residual does not increase, model $R^2$ will increase – we demonstrate this using simulated data (*Figure 5—figure supplement 1*). As the magnitude of the regression line's slope decreases, the regression line becomes a comparatively worse predictor compared to just using the mean, which decreases $R^2$ despite the mean error being the same across models. To demonstrate this in real data, we compared simple univariable models of log(BMI)~$PGS_{BMI}$ (in units of log(BMI)) between the bottom and top age quintiles in the European ancestry UKBB (*Figure 5—figure supplement 2*). As shown in previous sections, $R^2$ and $PGS_{BMI}$ beta

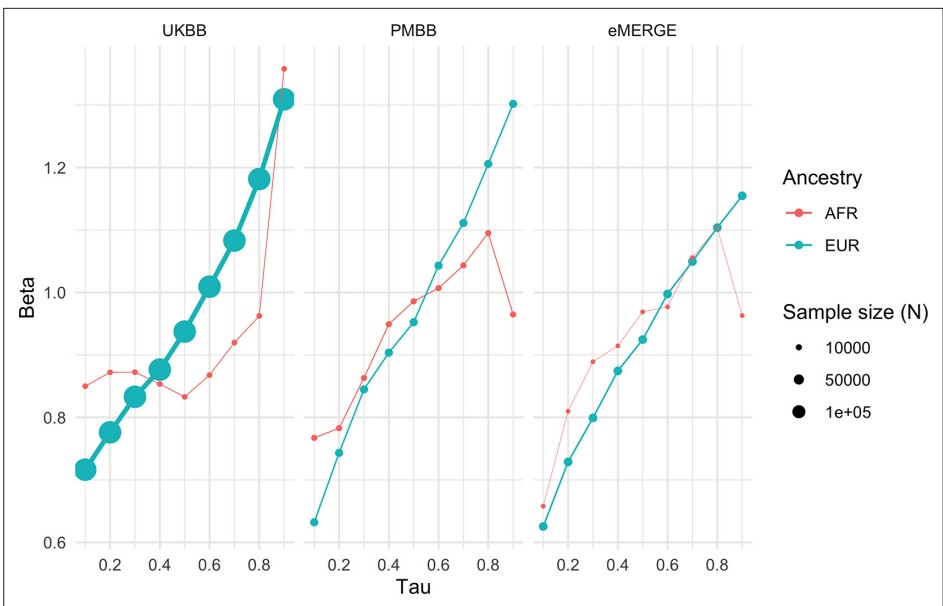

**Figure 5.** Quantile regression effects of PGS$_{BMI}$ (in units of log(BMI)) on log(BMI) at each decile of BMI in each cohort and ancestry. Tau is an input parameter for quantile regression corresponding to the percentile of the BMI distribution being modeled, with lower tau values representing the lower deciles (e.g., tau = 0.1 for the 10th percentile) and higher tau values representing the upper deciles (e.g., tau = 0.9 for the 90th percentile). The effect of PGS$_{BMI}$ increases as BMI itself increases, suggesting that no individual covariate–PGS interaction is responsible for the nonlinear effect of PGS$_{BMI}$. PGS, polygenic score; BMI, body mass index.

The online version of this article includes the following figure supplement(s) for figure 5:

**Figure supplement 1.** Three sets of simulated data with varying regression line slopes, showing how model $R^2$ changes when regression line slope changes, all else being equal.

**Figure supplement 2.** Univariable association of PGS$_{BMI}$ and log(BMI) in European UKBB, separately for the bottom and top quintiles of age.

are higher in younger individuals ($R^2$ = 0.088 versus $R^2$ = 0.066, and beta = 1.12 and 0.87, respectively), which seem to be a direct consequence of one another, as the mean squared error in younger individuals is actually higher (0.027 versus 0.022, respectively). This description suggests that the use of $R^2$ as the sole performance metric for evaluation of PGS may not always be appropriate, despite its overwhelming usage. Furthermore, this explanation helps explain the seemingly paradoxical results of significant interaction terms yet small increases in overall model $R^2$ and comparably much larger differences in $R^2$ in the stratified analyses.

## Effects of machine learning approaches on predictive performance

Given evidence of PGS–covariate dependence, we aimed to assess increases in $R^2$ when using machine learning models (neural networks), which can inherently model interactions and other nonlinearities, over linear models even with interaction terms. We first included age and sex as the only covariates (along with genotype PCs and PGS$_{BMI}$), as age and sex were present in all datasets and had significant and replicable evidence for PGS-dependence across our analyses. Three models were assessed – L1-regularized (i.e., LASSO) linear regression without any interaction terms, LASSO including a PGS–age and PGS–sex interaction term, and neural networks (without interaction terms). When comparing neural networks to LASSO with interaction terms, the relative tenfold cross-validated $R^2$ increased up to 67% (mean 23%) across cohorts and ancestries (*Figure 6*, *Supplementary file 1g*). The inclusion of interaction terms increased cross-validated $R^2$ up to 12% (mean 3.9%) when comparing LASSO including interaction terms to LASSO with main effects only.

We then modeled all available covariates and their interactions with PGS for each cohort and did similar comparisons. Cross-validated $R^2$ increased by up to 17.6% (mean 9.5%) when using neural networks versus LASSO with interaction terms, and up to 7.0% (mean 2.0%) when comparing LASSO

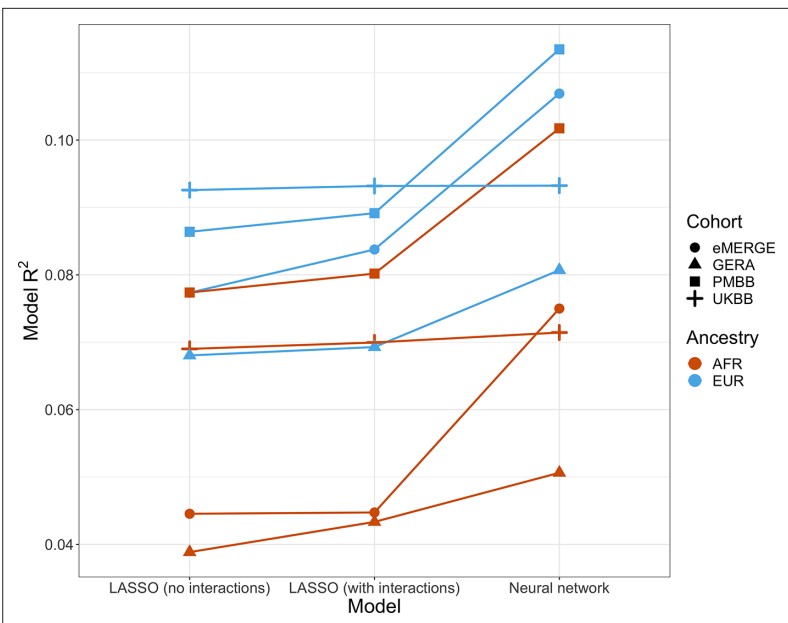

**Figure 6.** Model $R^2$ from different machine learning models across cohorts and ancestries using age and gender as covariates (along with $PGS_{BMI}$ and PCs 1–5). Across all cohorts and ancestries, LASSO with PGS–age and PGS– gender interaction terms had better average tenfold cross-validation $R^2$ than LASSO without interaction terms, while neural networks outperformed LASSO models. PGS, polygenic score; BMI, body mass index.

with interaction terms to LASSO with main effects only. Increases in model performance using neural networks were smaller in UKBB, perhaps due to the age range being smaller than other cohorts (all participants aged 39–73). This result suggests that additional variance explained through nonlinear effects with age and sex are explained by other variables present in the remainder of the datasets. Our findings show machine learning methods can improve model $R^2$ that include $PGS_{BMI}$ as variables beyond including interaction terms in linear models, even when variable selection is performed using LASSO, demonstrating that model performance can be increased beyond modeling nonlinearities through linear interaction terms and a feature selection procedure.

## Calculating PGS directly from GxAge GWAS effects

Previous studies *Mostafavi et al., 2020* have created PGS using GWAS stratified by different personal- level covariates, but for practical purposes this leads to a large loss of power as the full size of the GWAS is not utilized for each strata and continuous variables are forced into bins. We developed a novel strategy where PGS are instead created from a full-size GWAS that includes SNP–covariate interaction terms ('Materials and methods'). We focused on age interactions, given their large and replicable effects based on our results – similar to a previous study (*Mostafavi et al., 2020*), we conducted these analyses in the European UKBB. We used a 60% random split of study individuals to conduct three sets of PGS using GWAS of the following designs: main effects only, main effects also with an SNP–age interaction term, and main effects but stratified into quartiles by age. Twenty percent of the remaining data were used for training and the final 20% were held out as a test set. The best- performing PGS created from SNP–age interaction terms ($PGS_{GxAge}$) increased test $R^2$ to 0.0771 (95% bootstrap CI: 0.0770–0.0772) from 0.0715 (95% bootstrap CI: 0.0714–0.0716), a 7.8% relative increase compared to the best-performing main effect PGS (*Figure 7*, *Supplementary file 1h*) – age-stratified PGS had much lower performance than both other strategies (unsurprising given reasons previously mentioned). Including a $PGS_{GxAge}$–age interaction term only marginally increased $R^2$ (0.0001 increase), with similar increases for the other two sets of PGS, further demonstrating that post hoc modeling of interactions cannot reconcile performance gained through directly incorporating interaction effects from the original GWAS. The strategy of creating PGS directly from full-sized SNP–covariate interac- tions is potentially quite useful as it increases PGS performance without the need for additional data

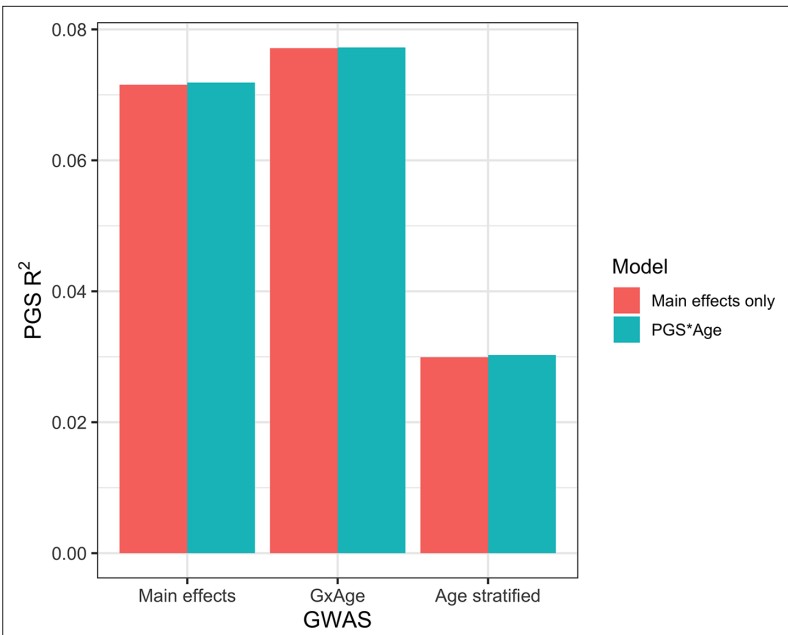

**Figure 7.** Polygenic score (PGS) $R^2$ based on three sets of genome-wide association studies (GWAS) setups. 'Main effects' were from a typical main effect GWAS, 'GxAge' effects were from a GWAS with an SNP–age interaction term, and 'Age stratified' GWAS had main effects only but were conducted in four age quartiles. PGS $R^2$ was evaluated using two models: one with main effects only and one with an additional PGS * Age interaction term.

– there are almost certainly a variety of points of improvement (described more in 'Discussion'), but we consider their investigation outside the scope of this study.

## Discussion

We uncovered replicable effects of covariates across four large-scale cohorts of diverse ancestry on both performance and effects of $PGS_{BMI}$. When stratifying by quintiles of different covariates, certain covariates had significant and replicable evidence for differences in $PGS_{BMI}$ $R^2$, with $R^2$ being nearly double between top- and bottom-performing quintiles for covariates with the largest differences. When testing PGS–covariate interaction effects, we also found covariates with significant interaction effects, where, for the largest effect covariates, each standard deviation change affected $PGS_{BMI}$ effect by nearly 20%. Across analyses, we found age and sex had the most replicable interaction effects, with levels of serum cholesterol, physical activity, and alcohol consumption having the largest effects across cohorts. Interaction effects and $R^2$ differences were strongly correlated, with main effects also being correlated with interaction effects and $R^2$ differences, suggesting that covariates with the largest interaction effects also contribute to the largest $R^2$ differences, with simple main effects also being predictive of expected differences in $R^2$ and interaction effects. Relatedly, we observed the effect of $PGS_{BMI}$ increases as BMI itself increases, and reason that differences in $R^2$ when stratifying by covariates are largely a consequence of difference in $PGS_{BMI}$ effects. Next, we employed machine learning methods for prediction of BMI with models that include $PGS_{BMI}$ and demonstrate that these methods outperform regularized linear regression models that include interaction effects. Finally, we employed a novel strategy that directly incorporates SNP interaction effects into PGS construction and demonstrate that this strategy improves PGS performance when modeling SNP–age interactions compared to PGS created only from main effects.

These observations are relevant to current research and clinical use of PGS, as individuals above a percentile cutoff are designated high risk (*Ge et al., 2019*), implying that individuals most at risk for obesity have both disproportionately higher predicted BMI and increased BMI prediction performance compared to the general population. More broadly, these results may likely extend to single variant effects instead of those aggregated into a PGS, which may inform the cause of previous GxE discoveries – for instance, variants near *FTO* that interact with physical activity discovered through

GWAS of BMI are robust and well-documented. However, individuals engaging in physical activity will generally have lower BMI than those that are sedentary, and these results suggest it may not be the difference in physical activity that is driving the interaction, rather the difference in BMI itself. This concept may also apply to other traits – for instance, sex-specific analyses are commonly performed, and variants with differing effects between male and female GWAS may largely be explained by phenotypic differences, rather than any combination of biological or lifestyle differences.

Future work may include replicating these analyses across additional traits, and trying to understand why these differences occur, as well as further exploring machine learning and deep learning methods on other phenotypes to determine if this trend of inclusion of PGS, along with covariate interaction effects, outperforms linear models for risk prediction. Additionally, inclusion of a PGS for the covariate to better measure its environmental effect is potentially worth exploring further and should improve in the future as PGS performance continues to increase. A slight limitation of this method in our study is that for the UKBB analyses the GWAS used for PGS construction were also from UKBB, thus not out-of-sample, although many of the covariates only have GWAS available through UKBB individuals. Furthermore, a variety of improvements are likely possible when creating PGS directly from SNP–covariate interaction terms. First, we used the same SNPs that were selected by pruning and thresholding based on their main effect p-values, but selection of SNPs based on their interaction p-values should also be possible and would likely improve performance. Additionally, performance of pruning and thresholding-based strategies has largely been overtaken by methods that first adjust all SNP effects for LD and do not require exclusion of SNPs, and a method that could do a similar adjustment for interaction effects would likely outperform most current methods for traits with significant context-specific effects. Next, incorporating additional SNP–covariate interactions (e.g., SNP–sex) would also likely further improve prediction performance, although any SNP selection/adjustment procedures may be further complicated by additional interaction terms. Finally, if SNP effects do truly differ according to differences in phenotype, then SNP effects would differ depending on the alleles one has, implying epistatic interactions are occurring at these SNPs.

While difference in phenotype itself may be able to explain the difference in genetic effects, it still may be that specific environmental or lifestyle characteristics are driving the differences. We propose several ideas about why BMI-associated covariates have larger interaction effects and impact on $R^2$ for $PGS_{BMI}$. First, age may be a proxy for accumulated gene–environment interactions as younger individuals have less exposure to environmental influences on weight compared to older individuals; therefore, one may expect that in younger individuals their phenotype could be better explained by genetics compared to environment. Second, PGS may more readily explain high phenotype values especially for positively skewed phenotypes, as large effect variants (e.g., associated with very high weight or height; *Robinson et al., 2006*) may be more responsible for extremely high phenotypic values. For example, the distribution of BMI is often positively skewed, and effects in trait-increasing alleles may have a larger potential to explain trait variation compared to trait-reducing variants. This explanation would likely be better suited to positively skewed traits and is not fully satisfactory as first log-transforming or rank-normal transforming the phenotype, as was done in this study, may invalidate this explanation.

PGS is a promising technique to stratify individuals for their risk of common, complex disease. To achieve more accurate predictions as well as promote equity, further research is required regarding PGS methods and assessments. This research provides firm evidence supporting the context-specific nature of PGS and the impact of nonlinear covariate effects for improving polygenic prediction of BMI, promoting equitable use of PGS across ancestries and cohorts.

## Materials and methods
### Study datasets
Individual inclusion criteria and sample sizes per cohort are described below – additional information is available in *Supplementary file 1a*.

### UKBB
Individual-level quality control (QC) and filtering have been described elsewhere (*Zhang et al., 2022*) for European ancestry individuals. Briefly, individuals were split by ancestry according to both

genetically inferred ancestry and self-reported ethnicity (*Bycroft et al., 2018*). Individuals with low genotyping quality and sex mismatch were removed, only unrelated individuals (Plink pi-hat < 0.250) were retained, and variants were filtered at INFO > 0.30 and minor allele frequency > 0.01. For African ancestry, individuals were first selected based on self-reported ethnicity 'Black or Black British', 'Caribbean', 'African', or 'Any other Black background'. Individuals who were low quality, that is, 'Outliers for heterozygosity or missing rate', and who were Caucasian from 'Genetic ethnic grouping' were removed. Of these individuals, those who were ±6 standard deviations from the mean of the first five genetic principal components provided by UKBB were excluded. Finally, only unrelated individuals were retained up to the second degree using plink2 (*Chang et al., 2015*) '-king-cutoff 0.125'. After QC and consideration of phenotype, a total of 7,046 individuals in the UKBB AFR data who also had BMI available were used for downstream analyses. In total, 383,775 individuals were used for analysis ($N_{EUR} = 376,729$, $N_{AFR} = 7,046$).

## eMERGE
Ancestry and relatedness inference have been described elsewhere (*Stanaway et al., 2019*). Individuals were split into European and African ancestry cohorts, and related individuals were removed (Plink pi-hat > 0.250) such that all were unrelated. In total, 35,064 individuals ($N_{EUR} = 31,961$, $N_{AFR} = 3,103$) were used for analysis.

## GERA
Ancestry inference has been described elsewhere (*Banda et al., 2015*), and study individuals were divided into European and African ancestry cohorts. Related individuals were removed using plink2 '-king-cutoff 0.125'. In total, 57,838 individuals ($N_{EUR} = 56,049$, $N_{AFR} = 1,789$) were used for analysis.

## PMBB
Ancestry inference and relatedness inference have been described elsewhere (*Penn Medicine BioBank, 2022*). Individuals were split into European and African ancestry cohorts, and related individuals were removed at pi-hat > 0.250. In total, 36,046 individuals ($N_{EUR} = 26,372$, $N_{AFR} = 9,674$) were used for analysis.

## Choice of covariates
A total of 62 covariates were included in the analyses, 25 of which were present (or similar proxies) in multiple datasets. These covariates were selected based on relevance to cardiometabolic health and obesity, and previous evidence of context-specific effects with BMI (*Rask-Andersen et al., 2017*; *Robinson et al., 2017*; *Justice et al., 2017*; *Tyrrell et al., 2017*; *Young et al., 2016*; *Winkler et al., 2015*). For UKBB, phenotype values were used from the collection that was closest to recruitment, and for PMBB the median values were used – for GERA and eMERGE, only one value was available. Additional details on covariate constructions, transformations, filtering, and cohort availability are provided in *Supplementary file 1b*.

## PGS construction
PGS for BMI ($PGS_{BMI}$) were constructed using PRS-CSx (*Ruan et al., 2022*), using GWAS summary statistics for individuals of European (*Locke et al., 2015*), African (*Ng et al., 2017*), and East Asian (*Sakaue et al., 2021*) ancestry that were out-of-sample of study participants. A set of 1.29 million HapMap3 SNPs provided by PRS-CSx was used for PGS calculation, which are generally well-imputed and variable frequency across global populations. Default settings for PRS-CSx (downloaded November 22, 2021) were used, which have been demonstrated to perform well for highly polygenic traits such as BMI (list of parameters is provided in *Supplementary file 1i*). The final $PGS_{BMI}$ per ancestry and cohort was calculated by regressing log(BMI) on the $PGS_{BMI}$ per ancestry without covariates – the combined, predicted value was used as a single $PGS_{BMI}$ in downstream analyses.

For GERA, BMI was not transformed as it was already binned into a categorical variable with five levels (18≤, 19–25, 26–29, 30–39, >40). Additionally, for GERA the uncombined ancestry-specific $PGS_{BMI}$ was used in the final models as it had higher $R^2$ than using the combined $PGS_{BMI}$ (data not shown).

## PGS_BMI performance after covariate stratification

Analyses were performed separately for each cohort and ancestry. For each covariate, individuals were binned by binary covariates or quintiles for continuous covariates. Incremental PGS_BMI $R^2$ was calculated by taking the difference in $R^2$ between:

$$\log(\text{BMI}) \sim \text{PGS}_{\text{BMI}} + \text{Age} + \text{Sex} + \text{PCs}_{1-5}$$

$$\log(\text{BMI}) \sim \text{Age} + \text{Sex} + \text{PCs}_{1-5}$$

We performed 5,000 bootstrap replications to obtain a bootstrapped distribution of $R^2$. p-Values for differences in $R^2$ were calculated between groups by calculating the proportion of overlap between two normal distributions of the $R^2$ value using the standard deviations of the bootstrap distributions. Again for GERA, BMI was not transformed.

## PGS_BMI interaction modeling

Evidence for interaction with each covariate with the PGS_BMI was evaluated using linear regression. It has been reported that the inclusion of covariates that are genetically correlated with the outcome can inflate test statistic estimates (*Aschard et al., 2015*; *Kerin and Marchini, 2020*; *Vanderweele et al., 2013*). To assuage these concerns, we introduced a novel correction, where we first calculated a PGS for the covariate (PGS_Covariate) and included it in the model, as well as an interaction term between PGS_BMI and PGS_Covariate. The PGS_Covariate terms were calculated using the European ancestry Neale Lab summary statistics (URLs) and PRS-CS (*Ge et al., 2019*). To standardize effect sizes across analyses, PGS_BMI and Covariate were first converted to mean zero and standard deviation of 1 (binary covariates were not standardized). We demonstrate inclusion of PGS_Covariate terms successfully reduced significance of the PGS_BMI * Covariate term (*Figure 3—figure supplement 1*). The final model used to evaluate PGS_BMI and Covariate interactions was

$$\log(\text{BMI}) \sim \text{PGS}_{\text{BMI}} * \text{Covariate} + \text{PGS}_{\text{BMI}} + \text{Covariate} + \text{PGS}_{\text{Covariate}} + \text{PGS}_{\text{BMI}} * \text{PGS}_{\text{Covariate}} +$$
$$\text{Age} + \text{Sex} + \text{PCs}_{1-5}$$

We report the effect estimates of the PGS_BMI * Covariate term, and differences in model $R^2$ with and without the PGS_BMI * Covariate term. Again for GERA, BMI was not transformed.

## Correlation between $R^2$ differences, interaction effects, and main effects

We estimated the main effects of each covariate on BMI with the following model:

$$\log(\text{BMI}) \sim \text{Covariate} + \text{Age} + \text{Sex} + \text{PCs}_{1-5}$$

Note that we ran new models with main effects only, instead of using the main effect from the interaction models (as the main effects in the interaction models depend on the interaction terms, and main effects used to create interaction terms are sensitive to centering of variables despite the scale invariance of linear regression itself; *Afshartous and Preston, 2011*). We then estimated the correlation between main effects, interaction effects, and maximum $R^2$ differences across all cohorts and ancestries weighting by sample size, analyzing quantitative and binary variables separately.

## Quantile regression to measure PGS effect across percentiles of BMI

The effect of PGS_BMI on BMI at different deciles of BMI was assessed using quantile regression. Tau – the parameter that sets which percentile to be predicted – was set to 0.10,0.20, …,0.90. Models included age, sex, and the top 5 genetic PCs as covariates. Analyses were stratified by ancestry and cohort, and BMI was first log transformed. GERA was excluded from these analyses as a portion of the models failed to run (as BMI values from GERA were already binned, some deciles all had the same BMI value – additionally, difference in effects between bins would be harder to evaluate as BMI within each decile would be more homogeneous).

## Machine learning models

UKBB EUR and GERA EUR models were restricted to 30,000 random individuals for computational reasons – BMI distributions did not differ from the full-sized datasets (Kolmogorov–Smirnov p-values of 0.29 and 0.57, respectively). $PGS_{BMI}$ and top 5 genetic principal components were included as features in all models. Two sets of models were evaluated for each cohort and ancestry: including age and sex as features, and including all available covariates in each cohort as features. Interaction terms between PGS and each covariate were included for models using interaction terms. 'Ever Smoker' status was used in favor of 'Never' versus 'Current smoking' status (if present) as individuals with 'Never' versus 'Current' status are a subset of those with 'Ever Smoker' status. UKBB AFR with all covariates was excluded due to small sample size (N = 53).

Neural networks were used as the model of choice, given their inherent ability to model interactions and other nonlinear dependencies. Prior to modeling, all features were scaled to be between 0 and 1. We used average tenfold cross-validation $R^2$ to evaluate model performance. Separate models were trained using untransformed and log(BMI). L1-regularized linear regression was used with 18 values of lambda ($1.0, 5.0 \times 10^{-1}, 2.0 \times 10^{-1}, 1.0 \times 10^{-1}, 5.0 \times 10^{-2}, 2.0 \times 10^{-2}, \ldots, 1.0 \times 10^{-5}, 5.0 \times 10^{-6}, 2.0 \times 10^{-6}$). Models were trained without inclusion of interaction terms (which neural networks can implicitly model) using 1,000 iterations of random search with the following hyperparameter ranges: size of hidden layers [10, 200], learning rate [0.01, 0.0001], type of learning rate [constant, inverse scaling], power t [0.4, 0.6], momentum [0.80, 1.0], batch size [32, 256], and number of hidden layers [1, 2].

## GxAge $PGS_{BMI}$ creation and assessment

Analyses were conducted in the European UKBB (N = 376,629), as was done in a study on a similar topic (*Mostafavi et al., 2020*). Three sets of analyses were performed using GWAS conducted in a 60% random split of individuals using the following models (BMI was rank-normal transformed before GWAS):

1. $\text{BMI} \sim \text{SNP} + \text{Age} + \text{Sex} + \text{PCs}_{1-5}$
2. $\text{BMI} \sim \text{SNP} + \text{Age} * \text{SNP} + \text{Age} + \text{Sex} + \text{PCs}_{1-5}$
3. Using the model in (1) but stratified into quartiles by age. BMI was rank-normal transformed within each quartile.

Using each set of GWAS, PGS was first calculated in a 20% randomly selected training set of the dataset using pruning and thresholding using 10 p-value thresholds ($0.50, 0.10, \ldots, 5.0 \times 10^{-5}, 1.0 \times 10^{-5}$) and remaining settings as default in Plink 1.9. For (2), GxAge $PGS_{BMI}$ was calculated using SNPs clumped by their main effect p-values from (1), and additionally incorporating the GxAge interaction effects per SNP. In other words, instead of typical PGS construction as

$$PGS_i = \beta_1 k_1 + \beta_2 k_2 + \ldots + \beta_n k_n$$

For an individual *i*'s PGS calculation, with main SNP effect β, and *n* SNPs, PGS incorporating GxAge effects ($PGS_{GxAge}$) was calculated as

$$PGS_{GXAge,i} = \beta_1 k_1 + \beta_{GXAge,1} k_1 Age_i + \beta_2 k_2 + \beta_{GXAge,2} k_2 Age_i \ldots + \beta_n k_n + \beta_{GXAge,n} k_n Age_i$$

where $\beta_{GxAge}$ is the GxAge effect for each SNP *n* and $Age_i$ is the age for individual *i*.

For each of the three analyses, the parameters and models resulting in the best-performing PGS (highest incremental $R^2$, using same main effect covariates as in the three GWAS) from the training set were evaluated in the remaining 20% of the study individuals. For (3), models were first trained within each quartile separately. To maintain sense of scale across quartiles (after rank-normal transformation), $R^2$ between all predicted values and true values was calculated together. For $R^2$ confidence intervals, the training set was bootstrapped and evaluated on the test set 5,000 times.

## URLs

Neale Lab UKBB summary statistics: http://www.nealelab.is/uk-biobank.

Select analysis code and data are available at https://github.com/RitchieLab/BMI_PGS_eLife (copy archived at *Ritchie Lab, 2024*).

## Acknowledgements

For UK Biobank: This research has been conducted using the UK Biobank Resource under Application Number 32133. This work uses data provided by patients and collected by the NHS as part of their care and support.

For GERA: Data came from a grant, the Resource for Genetic Epidemiology Research in Adult Health and Aging (RC2 AG033067; Schaefer and Risch, PIs) awarded to the Kaiser Permanente Research Program on Genes, Environment, and Health (RPGEH) and the UCSF Institute for Human Genetics. The RPGEH was supported by grants from the Robert Wood Johnson Foundation, the Wayne and Gladys Valley Foundation, the Ellison Medical Foundation, Kaiser Permanente Northern California, and the Kaiser Permanente National and Northern California Community Benefit Programs. The RPGEH and the Resource for Genetic Epidemiology Research in Adult Health and Aging are described in the following publication, Schaefer C, et al., The Kaiser Permanente Research Program on Genes, Environment and Health: Development of a Research Resource in a Multi-Ethnic Health Plan with Electronic Medical Records, In preparation, 2013.

For eMERGE: We acknowledge David Crosslin for helping clean the eMERGE data. Please see funding section for eMERGE funding acknowledgments.

For PMBB: We acknowledge the Penn Medicine BioBank (PMBB) for providing data and thank the patient- participants of Penn Medicine who consented to participate in this research program. We would also like to thank the Penn Medicine BioBank team and Regeneron Genetics Center for providing genetic variant data for analysis. The PMBB is approved under IRB protocol# 813913 and supported by Perelman School of Medicine at University of Pennsylvania, a gift from the Smilow family, and the National Center for Advancing Translational Sciences of the National Institutes of Health under CTSA award number UL1TR001878.

## Additional information

### Competing interests

Regeneron Genetics Center: Penn Medicine BioBank: The other authors declare that no competing interests exist.

### Funding

| Funder | Grant reference number | Author |
| --- | --- | --- |
| National Institutes of Health | AI077505 | Daniel Hui<br>Scott Dudek<br>Marilyn D Ritchie |
| National Institutes of Health | HL169458 | Daniel Hui<br>Scott Dudek<br>Marilyn D Ritchie |
| National Institute of Diabetes and Digestive and Kidney Diseases | DK52431 | Wendy K Chung |
| National Institutes of Health | U01 HG011166 | Josh F Peterson |
| National Institutes of Health | U01 HG008680 | Chunhua Weng |
| Group Health Cooperative/ University of Washington | U01HG008657 | Gail P Jarvik |

| Funder | Grant reference number | Author |
|---|---|---|
| Brigham and Women's Hospital | U01HG008685 | Elizabeth W Karlson |
| Vanderbilt University Medical Center | U01HG008672 | Wei-Qi Wei<br>Josh F Peterson<br>Qiping Feng |
| Cincinnati Children's Hospital Medical Center | U01HG008666 | Leah C Kottyan |
| Mayo Clinic | U01HG006379 | Iftikhar J Kullo<br>Johanna L Smith |
| Columbia University Health Sciences | U01HG008680 | Krzysztof Kiryluk<br>Atlas Khan<br>Chunhua Weng<br>Wendy K Chung |
| Northwestern University | U01HG008673 | Theresa L Walunas<br>Megan J Puckelwartz |
| Vanderbilt University Medical Center serving as the Coordinating Center | U01HG008701 | Josh F Peterson<br>Wei-Qi Wei |

The funders had no role in study design, data collection and interpretation, or the decision to submit the work for publication.

## Author contributions

Daniel Hui, Conceptualization, Formal analysis, Visualization, Methodology, Writing – original draft, Writing – review and editing; Scott Dudek, Formal analysis, Writing – review and editing; Krzysztof Kiryluk, Theresa L Walunas, Iftikhar J Kullo, Wei-Qi Wei, Hemant Tiwari, Josh F Peterson, Wendy K Chung, Brittney H Davis, Atlas Khan, Leah C Kottyan, Nita A Limdi, Qiping Feng, Megan J Puckelwartz, Chunhua Weng, Johanna L Smith, Elizabeth W Karlson, Gail P Jarvik, Writing – review and editing; Regeneron Genetics Center, DNA sequencing; Penn Medicine BioBank, Recruitment of participants, extraction of de-identified EHR data; Marylyn D Ritchie, Conceptualization, Resources, Supervision, Funding acquisition, Investigation, Methodology, Writing – original draft, Project administration, Writing – review and editing

## Author ORCIDs

Daniel Hui ⓘ http://orcid.org/0000-0002-8023-7352
Qiping Feng ⓘ https://orcid.org/0000-0002-6213-793X
Marylyn D Ritchie ⓘ https://orcid.org/0000-0002-1208-1720

Joint Public Review: https://doi.org/10.7554/eLife.88149.3.sa1
Author response https://doi.org/10.7554/eLife.88149.3.sa2

---

# Additional files

## Supplementary files

MDAR checklist

Supplementary file 1. Description. (a) Cohort descriptives. (b) All exposures and any transformations or filters. (c) $R^2$ differences across quintiles and binary variables. (d) Model descriptives on PRS–covariate interaction models. (e) Model descriptives on main effects. GERA correlations between main effects, interaction effects, and $R^2$ differences. Pearson R weighted by sample size are in bottom left, p-values in top right. (f) Pearson R weighted by sample size are in bottom left, p-values in top right. (g) Machine learning model descriptives. (h) Pruning and thresholding $R^2$ values and parameters for GxAge PGS. (i) Parameters used for PRS-CSx (default).

## Data availability

UK Biobank data was accessed under project #32133. eMERGE data is available at dbGaP in phs001584.v2.p2. GERA data is available at dbGaP in phs000674.v3.p3. PMBB individual level genotype and

phenotype data can be accessed through a research collaboration with a Penn investigator, as long as the requestor is from a not-for-profit organization. The PMBB genetic data used in this study were generated in collaboration with Regeneron Genetics Center; as such, we are unable to share the data with for-profit organizations without a three-way research collaboration agreement. Data sharing will require a PMBB project proposal and IRB approval. Collaboration requests can be sent to biobank@ upenn.edu. Select analysis code and data are available at https://github.com/RitchieLab/BMI_PGS_ eLife (copy archived at *Ritchie Lab, 2024*).

The following previously published datasets were used:

| Author(s) | Year | Dataset title | Dataset URL | Database and Identifier |
|---|---|---|---|---|
| Banda Y, Kvale MN, Hoffmann TJ, Hesselson SE, Ranatunga D, Tang H | 2015 | Resource for Genetic Epidemiology Research on Aging (GERA) | https://www.ncbi.nlm. nih.gov/projects/gap/ cgi-bin/study.cgi? study_id=phs000674. v3.p3 | dbGaP, phs000674.v3.p3 |
| Stanaway IB, Hall TO, Rosenthal EA, Palmer M, Naranbhai V, Knevel R | 2020 | eMERGE Network Phase III: HRC SNV and 1000 Genomes SV Imputed Array Data of 105,000 Participants | https://www.ncbi.nlm. nih.gov/projects/gap/ cgi-bin/study.cgi? study_id=phs001584. v2.p2 | dbGaP, phs001584.v2.p2 |

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

## Appendix 1

### Regeneron Genetics Center (RGC) banner author list and contribution statements

### RGC management and leadership team

Goncalo Abecasis, PhD, Aris Baras, M.D., Michael Cantor, M.D., Giovanni Coppola, M.D., Andrew Deubler, Aris Economides, Ph.D., Luca A. Lotta, M.D., Ph.D., John D. Overton, Ph.D., Jeffrey G. Reid, Ph.D., Katherine Siminovitch, M.D., Alan Shuldiner, M.D.

### Sequencing and lab operations

Christina Beechert, Caitlin Forsythe, M.S., Erin D. Fuller, Zhenhua Gu, M.S., Michael Lattari, Alexander Lopez, M.S., John D. Overton, Ph.D., Maria Sotiropoulos Padilla, M.S., Manasi Pradhan, M.S., Kia Manoochehri, B.S., Thomas D. Schleicher, M.S., Louis Widom, Sarah E. Wolf, M.S., Ricardo H. Ulloa, B.S.

### Clinical informatics

Amelia Averitt, Ph.D., Nilanjana Banerjee, Ph.D., Michael Cantor, M.D., Dadong Li, Ph.D., Sameer Malhotra, M.D., Deepika Sharma, MHI, Jeffrey Staples, Ph.D.

### Genome informatics

Xiaodong Bai, Ph.D., Suganthi Balasubramanian, Ph.D., Suying Bao, Ph.D., Boris Boutkov, Ph.D., Siying Chen, Ph.D., Gisu Eom, B.S., Lukas Habegger, Ph.D., Alicia Hawes, B.S., Shareef Khalid, Olga Krasheninina, M.S., Rouel Lanche, B.S., Adam J. Mansfield, B.A., Evan K. Maxwell, Ph.D., George Mitra, B.A., Mona Nafde, M.S., Sean O'Keeffe, Ph.D., Max Orelus, B.B.A., Razvan Panea, Ph.D., Tommy Polanco, B.A., Ayesha Rasool, M.S., Jeffrey G. Reid, Ph.D., William Salerno, Ph.D., Jeffrey C. Staples, Ph.D., Kathie Sun, Ph.D.

### Analytical genomics and data science

Goncalo Abecasis, D.Phil., Joshua Backman, Ph.D., Amy Damask, Ph.D., Lee Dobbyn, Ph.D., Manuel Allen Revez Ferreira, Ph.D., Arkopravo Ghosh, M.S., Christopher Gillies, Ph.D., Lauren Gurski, B.S., Eric Jorgenson, Ph.D., Hyun Min Kang, Ph.D., Michael Kessler, Ph.D., Jack Kosmicki, Ph.D., Alexander Li, Ph.D., Nan Lin, Ph.D., Daren Liu, M.S., Adam Locke, Ph.D., Jonathan Marchini, Ph.D., Anthony Marcketta, M.S., Joelle Mbatchou, Ph.D., Arden Moscati, Ph.D., Charles Paulding, Ph.D., Carlo Sidore, Ph.D., Eli Stahl, Ph.D., Kyoko Watanabe, Ph.D., Bin Ye, Ph.D., Blair Zhang, Ph.D., Andrey Ziyatdinov, Ph.D.

### Therapeutic area genetics

Ariane Ayer, B.S., Aysegul Guvenek, Ph.D., George Hindy, Ph.D., Giovanni Coppola, M.D., Jan Freudenberg, M.D., Jonas Bovijn M.D., Katherine Siminovitch, M.D., Kavita Praveen, Ph.D., Luca A. Lotta, M.D., Manav Kapoor, Ph.D., Mary Haas, Ph.D., Moeen Riaz, Ph.D., Niek Verweij, Ph.D., Olukayode Sosina, Ph.D., Parsa Akbari, Ph.D., Priyanka Nakka, Ph.D., Sahar Gelfman, Ph.D., Sujit Gokhale, B.E., Tanima De, Ph.D., Veera Rajagopal, Ph.D., Alan Shuldiner, M.D., Bin Ye, Ph.D., Gannie Tzoneva, Ph.D., Juan Rodriguez-Flores, Ph.D.

### Research program management and strategic initiatives

Esteban Chen, M.S., Marcus B. Jones, Ph.D., Michelle G. LeBlanc, Ph.D., Jason Mighty, Ph.D., Lyndon J. Mitnaul, Ph.D., Nirupama Nishtala, Ph.D., Nadia Rana, Ph.D., Jaimee Hernandez

### Penn Medicine BioBank Banner Author List and Contribution Statements

### PMBB Leadership Team

Daniel J. Rader, M.D., Marylyn D. Ritchie, Ph.D.

Contribution: All authors contributed to securing funding, study design and oversight. All authors reviewed the final version of the manuscript.

## Patient Recruitment and Regulatory Oversight

JoEllen Weaver, Nawar Naseer, Ph.D., M.P.H., Giorgio Sirugo, M.D., P.h.D., Afiya Poindexter, Yi-An Ko, Ph.D., Kyle P. Nerz

Contributions: JW manages patient recruitment and regulatory oversight of study. NN manages participant engagement, assists with regulatory oversight, and researcher access. GS assists with researcher access. AP, YK, KPN perform recruitment and enrollment of study participants.

## Lab Operations

JoEllen Weaver, Meghan Livingstone, Fred Vadivieso, Stephanie DerOhannessian, Teo Tran, Julia Stephanowski, Salma Santos, Ned Haubein, P.h.D., Joseph Dunn

Contribution: JW, ML, FV, SD conduct oversight of lab operations. ML, FV, AK, SD, TT,JS, SS perform sample processing. NH, JD are responsible for sample tracking and the laboratory information management system.

## Clinical Informatics

Anurag Verma, Ph.D., Colleen Morse Kripke, M.S. DPT, MSA, Marjorie Risman, M.S., Renae Judy, B.S., Colin Wollack, M.S.

Contribution: All authors contributed to the development and validation of clinical phenotypes used to identify study subjects and (when applicable) controls.

## Genome Informatics

Anurag Verma Ph.D., Shefali S. Verma, Ph.D., Scott Damrauer, M.D., Yuki Bradford, M.S., Scott Dudek, M.S., Theodore Drivas, M.D., Ph.D.

Contribution: AV, SSV, and SD are responsible for the analysis, design, and infrastructure needed to quality control genotype and exome data. YB performs the analysis. TD and AV provides variant and gene annotations and their functional interpretation of variants.

For PMBB, please use: biobank@pennmedicine.upenn.edu

For Regeneron, please use: RGCcollaborations@regeneron.com

