## [Editor Report · eLife assessment]

This study presents a **convincing** analysis of the effects of covariates, such as age, sex, socioeconomic status, or biomarker levels, on the predictive accuracy of polygenic scores for body mass index; the work is further supported by **important** approaches for improving prediction accuracy by accounting for such covariates across a variety of association studies. The authors did a commendable job addressing reviewer suggestions and comments. The work will be of interest to colleagues using and developing methods for phenotypic prediction based on polygenic scores.

---

## [Referee Report · Joint Public Review]

In this paper Hui and colleagues investigate how the predictive accuracy of a polygenic score (PGS) for body mass index (BMI) changes when individuals are stratified by 62 different covariates. After showing that the PGS has different predictive power across strata for 18 out of 62 covariates, they turn to understanding why these differences and seeing if predictive performance could be improved. First they investigated which types of covariates result in the largest differences in PGS predictive power, finding that covariates with with larger "main effects" on the trait and covariates with larger interaction effects (interacting with the PGS to affect the trait) tend to better stratify individuals by PGS performance. The authors then see if including interactions between the PGS and covariates improves predictive accuracy, finding that linear models only result in modest increases in performance but nonlinear models result in more substantial performance gains.

Overall, the results are interesting and well-supported. The results will be broadly interesting to people using and developing PGS methods, as well as the broader statistical genetics community.

A few of the important points of the paper are:

A major impediment to the clinical use of PGS is the interaction between the PGS and various other routinely measure covariates, and this work provides a very interesting empirical study along these lines. The problem is interesting, and the work presented here is a convincing empirical study of the problem.

The result that PGS accuracy differs across covariates, but in a way that is not well-captured by linear models with interactions is important for PGS method development.

The quantile regression analysis is an interesting approach to explore how and why PGS may differ in accuracy across different strata of individuals.

---

## [Author Response]

The following is the authors’ response to the original reviews.

We previously responded to reviewer comments in a previous iteration of this draft, edited the manuscript accordingly, and have no further comments on the majority of them. However, we performed additional analyses mainly in response to weaknesses Reviewer 1 highlighted related to “one shortcoming [being] the lack of a conceptual model explaining the results”, and the eLife assessment stating “the study falls short of providing a cogent interpretation of key findings, which could be of great interest and utility”. We provide a conceptual explanation that ties together many of our results, which we demonstrate using real data and further explore using simulated data – these analyses are in a new section titled “Increase in PGS effect for increasing percentiles of BMI itself, and its relation to R2 differences when stratifying by covariates”, with the Discussion also being updated accordingly.

Essentially, we demonstrate that the effect of PGSBMI increases as BMI itself increases (using quantile regression – newly created Figure 5). This finding helps explain the correlation between covariate main effects, interaction effects, and maximum R2 differences when stratifying on different covariates, and also why any one or combination of covariates did not seem to be of unusual interest. While this result readily explains why covariates with larger main effects have larger interaction effects, by itself it does not seem to explain the differences in R2 in covariate-stratified bins, but we show using portions of real data and simulated data that in the case of this study they are closely related.

Effectively, as the effect of PGSBMI increases, variance in the phenotype will also increase – so long as the residuals do not increase proportionately, this causes R2 to also increase as R2 directly depends on outcome variance. We demonstrate this using simulated data (newly created S Figure 2) and real data (newly created S Figure 3). So the largest R2 differences between certain covariate-stratified bins seems to be a direct consequence of those covariates also having the largest PGSBMI*covariate interaction effects. These results tie into our previous response to Reviewer 1, where essentially there is not only heteroskedasticity in the relationship between PGSBMI and BMI, but a cause of the heteroskedasticity is an increasing effect in PGSBMI as BMI itself increases.

In the Discussion, we highlight several broad implications of these findings. First, these results may, in part, provide a generalizable explanation for epistasis, as the effect of a PGS (or any individual SNP) seems to depend on phenotype, and as phenotype depends on many SNPs, the effect of PGS and individual SNPs depends on other SNPs. Second, these results may also provide a generalizable explanation for GxE, as, demonstrated in this paper, interaction effects for SNPs (or a PGS) may largely depend on the phenotypic value itself, rather than any specific environment(s) or combination of. Finally, related to our previous response to Reviewer 2, modeling effects of SNPs dependent on phenotype itself would almost certainly result in gains in PGS performance (and locus discovery), which should also be larger than e.g., just GxAge effects as we demonstrated in this manuscript.